# Exploring *Xenorhabdus* and *Photorhabdus* Nematode Symbionts in Search of Novel Therapeutics

**DOI:** 10.3390/molecules29215151

**Published:** 2024-10-31

**Authors:** Ewa Sajnaga, Waldemar Kazimierczak, Magdalena Anna Karaś, Monika Elżbieta Jach

**Affiliations:** 1Department of Biomedicine and Environmental Research, John Paul II Catholic University of Lublin, Konstantynów 1J, 20-708 Lublin, Poland; waldemar.kazimierczak@kul.pl; 2Department of Genetics and Microbiology, Institute of Biological Science, Faculty of Biology and Biotechnology, Maria Curie-Skłodowska University, Akademicka 19, 20-033 Lublin, Poland; magdalena.karas@mail.umcs.pl; 3Department of Molecular Biology, John Paul II Catholic University of Lublin, Konstantynów 1H, 20-708 Lublin, Poland; monika.jach@kul.pl

**Keywords:** *Xenorhabdus*, *Photorhabdus*, natural products, bioactive compounds, antibiotics, drug leads, genome mining, entomopathogenic nematodes

## Abstract

*Xenorhabdus* and *Photorhabdus* bacteria, which live in mutualistic symbiosis with entomopathogenic nematodes, are currently recognised as an important source of bioactive compounds. During their extraordinary life cycle, these bacteria are capable of fine regulation of mutualism and pathogenesis towards two different hosts, a nematode and a wide range of insect species, respectively. Consequently, survival in a specific ecological niche favours the richness of biosynthetic gene clusters and respective metabolites with a specific structure and function, providing templates for uncovering new agrochemicals and therapeutics. To date, numerous studies have been published on the genetic ability of *Xenorhabdus* and *Photorhabdus* bacteria to produce biosynthetic novelty as well as distinctive classes of their metabolites with their activity and mechanism of action. Research shows diverse techniques and approaches that can lead to the discovery of new natural products, such as extract-based analysis, genetic engineering, and genomics linked with metabolomics. Importantly, the exploration of members of the *Xenorhabdus* and *Photorhabdus* genera has led to encouraging developments in compounds that exhibit pharmaceutically important properties, including antibiotics that act against Gram- bacteria, which are extremely difficult to find. This article focuses on recent advances in the discovery of natural products derived from these nematophilic bacteria, with special attention paid to new valuable leads for therapeutics.

## 1. Introduction

Natural products (NPs) with low molecular weight, also called specialised or secondary metabolites, released by living organisms play an important ecological role by exerting effects on both producers and neighbouring organisms. Through specific binding to cellular targets, they can, e.g., defend against enemies, facilitate communication within the population, or fend off competing organisms. The NP structures range from simple to very complex, indicating their extraordinary adaptability to different environmental conditions [1]. This huge diversity of chemical scaffolds is an exceptionally favourable feature of NPs as drug candidates, especially since natural precursors may undergo further synthetic modification or reduction in size and chirality while still maintaining their bioactivity to better meet pharmaceutical requirements. In 1940–1960, called ’the golden age of discovery’, NPs from a few taxa of easy cultured microorganisms, especially *Streptomyces*, *Bacillus*, and *Pseudomonas*, were the most important sources of pharmaceuticals, especially for the treatment of infectious, cardiovascular, and cancer diseases [2]. However, over the years, the frequency of new NP discoveries decreased significantly due to several limitations, such as the time-consuming nature and unreliability of bioassays, difficulties in the separation of NPs and their instability, a high rediscovery rate, or difficulty in the optimisation of production. For this reason, pharmaceutical companies have reduced investment in NP research, especially in the United States, perceiving it as a slow process and assuming that all easy NPs have already been discovered [1]. However, with the advent of automated clinical analysers and progress in next-generation sequencing (NGS) that addresses many of the previous limitations, NP research is currently attracting the attention of scientists and the pharmaceutical industry again, in hopes of tackling the antimicrobial resistance crisis and the rash of oncological diseases [3,4,5]. In addition, to avoid identification of nonspecific NPs in overmined typical producers, there is a strong need to look for new promising sources of specialised metabolites. Great hopes are aroused by studies of microbes from soil and marine environments, including previously uncultured bacteria, which can grow using modern platforms, for example, iChip [6]. This approach has resulted in the discovery of new promising antibiotic teixobactins originating from soil β-proteobacterium *Eleftheria terra*, which is unculturable with traditional methods [7]. Of particular interest are also microorganisms engaged in close interaction with animals, e.g., members of the *Burkholderia* genus, comprising both pathogenic and symbiotic species [8,9]. Accordingly, many efforts are aimed at exploring *Xenorhabdus* and *Photorhabdus* (XP) bacteria, which, as obligate gut symbionts of entomopathogenic nematodes (EPNs), kill the insect and later protect its cadaver from competitors with a broad range of bioactive molecules, which are not toxic to the symbiotic host. During their life cycle, XB bacteria interact with members of the microbiota of both nematodes and insects, varying according to the type of individuals and its geographical origin. The bacterial community associated with EPNs has been shown to consist of Gram- *Acinetobacter*, *Stenotrophomonas*, *Pseudomonas*, *Alcaligenes*, and *Providencia*, and some of these bacteria can be released by the host together with its mutualists into insect hemolymph during infection, probably contributing to nematode virulence [10,11,12]. In turn, soil-dwelling coleopteran larvae, such as *Melolontha* spp., which are frequently infected by EPNs, have a gut microbiota dominated by γ-proteobacteria, such as *Enterobacteriaceae* or *Pseudomonadaceae* [13,14]. These bacteria can also translocate into the insect hemolymph through intestinal perforations caused by infective juveniles (IJs) of the nematodes and then share the same compartment; therefore, released antimicrobials are crucial in competitive events that occur during the early and late stages of infection [15,16].

## 2. *Xenorhabdus* and *Photorhabdus* Bacteria as Nematode Symbionts

Bacteria from the genera *Xenorhabdus* and *Photorhabdus* (γ-proteobacteria: *Morganellaceae*) engage in an obligate mutualistic relationship with cosmopolite EPNs of the genera *Steinernema* and *Heterorhabditis*, respectively, being highly pathogenic in this combination to a wide range of soil insects [17,18]. Phylogenetic analyses have confirmed that these two genera are closely related; however, each of them is highly diverse, with many species and subspecies still being identified [19,20]. Currently, the genera *Xenorhabdus* and *Photorhabdus* contain 30 and 23 species, respectively, and some include subspecies (Figure 1). However, the complete picture of the biodiversity of these bacteria remains limited, especially since symbiotic bacteria from many species of EPN have not yet been described [21].

Generally, all XP bacteria show a similar lifestyle comprising mutualistic and pathogenic stages of life. In brief, the bacteria are housed in the intestine of IJs, which is the only free-living stage of nematodes. IJs invade insect larvae living in the soil, and release the bacteria into the hemolymph. Then, the bacteria start to propagate and produce a mixture of protein toxins, hydrolases, and NPs suppressing the insect immune response, which leads to the fast death of the larvae. Within the nutritional insect cadaver, the bacteria continue to grow and produce metabolites to ensure the degradation of insect tissue, development of nematode progeny, suppression of microbial competitors, and deterrence of opposing nematodes and saprophytic scavengers [22,23,24]. After several reproductive cycles of nematodes when the food resources of the insect cadaver are depleted, a new generation of IJs reassociate with bacterial mutualists and emerge to initiate a new infection cycle [25]. Generally, each nematode species associates with specific *Xenorhabdus* or *Photorhabdus* bacteria; however, bacterial species can have multiple nematode hosts [26]. A symbiont switching experiment has demonstrated decreasing virulence and reproduction of nematodes after associating with non-cognitive symbionts at the subspecies level [27,28]. The molecular basis of these complex relationships has not been well understood yet; however, studies have revealed a complex cascade of regulators and inter-kingdom signalling molecules, with the same molecular mechanisms that overlap between pathogenicity and mutualism, for example, those involved in LPS production [29,30]. Phylogenetic studies have shown that the virulence of XP bacteria against insects has generally increased during the course of evolution [31], although avirulent or virulence-attenuated strains injected alone have also been found [32,33]. Due to their strong entomopathogenic properties, high specificity to target pests, and non-toxicity to humans, EPN complexes with XP bacteria have been used for many years as biopesticides, representing a promising approach in sustainable agriculture [34].

Interestingly, XP bacteria can occur in two phenotypic cell variants, i.e., primary (1°) and secondary (2°), differing in several traits, including faster growth and limited production of secondary metabolites, which results in lower pathogenicity of the latter toward insects. While only 1° cells enter symbiosis with EPNs and are highly pathogenic to insects, the role of the 2° form is unclear [35]. However, research has demonstrated that 2° cells of *Photorhabdus luminescens* that remain in the soil are capable of specific interactions with plant roots and protection thereof from phytopathogenic fungi [36]. Therefore, further studies of XP bacteria may open the door to future applications of these bacteria not only as biopesticides in agriculture but also as plant-protecting agents.

## 3. Natural Products of *Xenorhabdus* and *Photorhabdus* Bacteria

Numerous NPs regulated by complex signal pathways help to maintain the tripartite bacteria–nematode–insect interaction during the complex life cycle of XP bacteria [37]. The exploration of XP in search of novel NPs began in the early 1980s. It was noticed that the dead insects infected by EPNs did not decay, suggesting the production of antimicrobials by bacterial symbionts. This hypothesis was tested by in vitro antibiotic assays in a study conducted by Ackhurst (1982), who revealed that only 1° cells of *Xenorhabdus* spp. inhibited a wide range of both Gram+ and Gram- bacteria and yeasts [38]. The first antibacterial compound isolated and structurally elucidated was stilbene from *Photorhabdus* [35,39]. Subsequently, in vivo studies confirmed the presence of multiple antibiotics in infected insect cadavers [40]. Before 2000, several other NPs with antibiotic activity had been identified, including indole and anthraquinone derivatives [41,42]. The publication of the first genome sequence of *P. luminescens* TT01 revealed that up to 6% of the genome is dedicated to secondary metabolism, confirming the importance of these bacteria as a source of new bioactive compounds [43]. Later, due to the application of modern mass spectrometry (MS) and molecular methods, research accelerated and dozens of newly identified biosynthetic gene clusters (BGCs) and respective products were reported [44]. The most widely examined are prevalent nonribosomal peptides, especially as this easy-to-manipulate class of NPs is the main source of commercial antibiotics [45]. Many tested NPs, for example, benzaldehyde, indole, anthraquinones, fabclavines, isopropylstilbenes, xenocoumacins, or xenorhabdins, which are universal protection factors, have a broad spectrum of activity, while others serve specific functions, targeting specific insect immune pathways [46,47] (Figure 2, Table 1).

An undoubted advantage of XP bacteria in terms of uncovering new chemical scaffolds is the ease and low cost of their cultivation. Isolated from IJs or infected insects, XP bacteria can grow in standard media, and all their life stages can be maintained in the laboratory, which is important because many NPs are not produced when the organism lacks a natural habitat. The biological effects of NPs released by XP bacteria have been evaluated in standard laboratory conditions using two animal models, i.e., IJs and insects (usually *Galleria mellonella*, Lepidoptera: Pyralidae), in terms of mutualistic recovery and pathogenicity, respectively. The studies revealed that NPs are crucial for the colonisation of mutualistic bacteria in specific nematode hosts, communication with other organisms, pathogenesis, and protection of the host cadaver against competitors [48]; however, the biological function of many NPs is still elusive. Exploration of NPs in their native environment helps decipher the ecological function of discovered metabolites, which in turn accelerates drug discovery. For example, because of functional conservation, detected analogues may play a similar pharmaceutically relevant function, but with modulated properties.

**Table 1 molecules-29-05151-t001:** List of specialised metabolites with relative BGCs detected both in *Xenorhabdus* and *Photorhabdus* bacteria, their bioactivity, and biological functions.

BGC Product	Class/ Biosynthetic Pathway *	Bioactivity	Biological Function/ Mechanism of Action	References
β-lactone	β-lactone	Insecticidal, immunosuppressive	Proteosome inhibitor disturbing cell cycle and causing immunodeficiency	[49]
GameXPeptides	NRPS	Insecticidal, immunosuppressive, antiprotozoal	Suppressing nodule formation and spread of haemocytes. Unknown target, possibly heat shock proteins	[49,50,51]
Odilorhabdin	NRPS	Antibacterial	By binding to the small ribosome subunit, it induces miscoding during translation	[52,53,54]
Photoxenobactin	PKS/NRPS hybrid	Insecticidal	Virulence-related siderophore	[49]
Phurealipid	Urealipid	Insecticidal, immunosuppressive	Mimics juvenile hormone to suppress insect immunity and immature development. Suppresses the production of antimicrobial peptides	[51,55,56]
Putrebactin/avoroferrin	Siderophore	Insecticidal, immunosuppressive	Repression of histone deacetylase, suppressing the production of antimicrobial peptides	[57]
Pyrazinone/lumizinone	NRPS	Cytotoxic	Possibly a cysteine protease inhibitor, disrupting the activation of signalling pathways	[58]
Pyrrolizixenamide	NRPS	Antibacterial, antitumor, immunosuppressive	Probably a phospholipase A2 inhibitor	[59]
Rhabdobranin	PKS/NRPS hybrid	Unknown	Prodrug activity mechanism	[49]
Rhabdopeptide/ xenortide peptide	NRPS	Insecticidal, immunosuppressive, antiprotozoal, nematocidal, cytotoxic	Possibly inhibiting the serine protease cascade by preventing prophenyloxidase activation	[60,61,62,63,64]
Rhabduscin	Other	Insecticidal, immunosuppressive	Inhibition of phenoloxidase	[65]
Xenematide	NRPS	Antibacterial, insecticidal	Unknown	[66]
Xenorhabdin/ xenorxide	NRPS	Insecticidal, antibacterial, antifungal, antiprotozoal, cytotoxic	Proteasome inhibitor Inhibition of RNA synthesis	[67,68]

* NPRS—nonribosomal peptide synthetase; PKS—polyketide synthetase.

### 3.1. Natural Products Derived from Xenorhabdus and Photorhabdus Bacteria Targeting Eukaryotic Cells

XP bacteria are generally best known as an important source of virulence factors, i.e., toxins, small proteins, and natural products. They are investigated for use in agriculture as innovative bioinsecticides, independently or in combination with commercial *Bacillus thuringiensis* to improve plant protection [44,69,70]. However, in terms of development of therapeutic agents, NPs with immunosuppressive activity are receiving more attention, as they are needed to treat autoimmune diseases and prevent transplanted tissue and organ rejection. In fact, numerous virulent NPs secreted by XP bacteria into the insect hemocoel specifically target several aspects of the insect immune system to suppress it and induce fatal septicaemia or modulate the immune response of EPNs, leading to stable symbiosis [71,72]. To cause immunosuppression, XP bacteria commonly inhibit phospholipase A2 (PLA2), which is involved in the biosynthesis of immune-mediating eicosanoids. Studies have revealed several compounds with anti-PLA2 activity, such as the most efficient benzylideneacetone (BZA), indole, oxindole, and *p*-hydroxyphenyl propionic acid [73,74,75]. Another target for immunosuppressive metabolites may be phenoloxidases (PO), i.e., key enzymes of insect immunity responsible for producing melanin, a polymer that seals off microbial pathogens. A well-known PO inhibitor is the tyrosine derivative rhabduscin, which is associated with the surface of the bacterial cell and provides protection against host defences [65]. Other known PO inhibitors include 4-hydroxystilbene, benzylideneacetone, 1,2-benzenedicarboxylic acid, benzaldehyde, and oxindoles. Immunosuppressive metabolites are produced sequentially during different growth phases and cooperatively inhibit different key elements of insect immune responses [76]. Interestingly, in the case of *Photorhabdus asymbiotica*, which is capable of evading mammalian tissues, significantly increased production of rhabduscin facilitates protection against death by the humoral immunity cascade [77]. Other immunosuppressive NPs can interrupt the serine protease cascade (rhabdopeptides, xenortides), the nitric oxide pathway (lipocitides), and the proteasome machinery (glidobactin A, xenorhabdin) or suppress the production of antimicrobial peptides (putrebactin, phurealipids) [47].

Biocontrol of insects transmitting arboviruses, which cause several tropical diseases, e.g., dengue, chikungunya, or yellow fever, is regarded as an efficient and eco-friendly method to reduce the spread of infections, alternative to chemical products. Numerous studies of XP bacteria that have been carried out so far have shown larvicidal properties, many of them acting against mosquitoes with medical importance, such as *Aedes*, *Culex*, and *Anopheles* spp. [78]. For example, *Xenorhabdus nematophila* culture broth as well as its mixture with *B. thuringiensis* subsp. *israelensis* showed high toxicity against culicides [79,80]. Other studies focus on *Xenorhabdus innexi*, whose culture fluids showed high larvicidal activity against mosquitoes after ingestion but not against non-target lepidopteran insects. The active compound was described as a lipopeptide toxin and was further identified as a member of the fabclavine family [81]. Research has also demonstrated the larvicidal potential of anthraquinones, xenocoumacins, and the PirAB protein [82,83,84]. 

In addition, conventional acaricides used currently to control mite pests, e.g., *Tetranychus urticae*, raise environmental and developing resistance concerns; therefore, it is necessary to search for new acaricidal biopesticides. Studies have indicated that NPs derived from *Xenorhabdus* bacteria are efficient in killing mites; however, only one acaricidal substance, xenocoumacin, has been identified so far [85]. Other important NPs from XP bacteria are compounds showing nematistatic activity, including fabclavines, rhabdopeptides, xenocoumacins, and trans-cinnamic acid; however, the knowledge of their mechanisms of action is still in its infancy [64,86]. However, this group of compounds have potential to be used against plant parasitic nematodes representing a global problem in agriculture and as anthelmintic drugs controlling human and animal nematode infection. 

Many natural products released by XP bacteria exhibit cytotoxic properties toward mammalian cells [64], but few have pharmaceutically desirable specificity towards cancer cells. For example, antiprotozoal xenortides discovered in *Xenorhabdus* have strong anticancer activity and are not toxic to normal cells [87]. Furthermore, *Xenorhabdus stockiae* exhibited significant cytotoxic activity against human epithelial carcinoma cells and prevented lung metastases [88]. Known for conferring cytotoxic bioactivity is also the benzoxazolinate moiety with a rare bis-heterocyclic structure that enables interaction with both proteins and DNA. The best-known benzoxazolinate-containing NP is lidamycin (C-1027), which shows activity against Gram+ bacteria and cytotoxicity toward cancer cell lines through induction of DNA double-strand breaks [89]. Benzobactin, i.e., a representative of this class of NPs exhibiting cytotoxic properties, was previously identified in *Xenorhabdus vietnamensis* [49]. Recently, Shi et al. (2022) have characterised benzoxazolinate-containing metabolites using the genome mining of bacterial genomes collected in the NCBI database [90]. They have shown that BGCs for various benzoxazolinates are spread not only in XP bacteria but also across diverse groups of bacteria in the Proteobacteria and Firmicute phyla, which is an advantageous feature for harnessing this huge biosynthetic potential for pharmaceutical applications. Relative cepafungins, glidobactins, and luminmycins have also been reported to have strong cytotoxic activity against human cancer cells [91,92]. The mechanism responsible for this activity is the inhibition of the proteasome, which is the main element of the ubiquitin-based protein degradation system in all eukaryotes, essential for the regulation of cellular processes. Thus, the inhibition of the proteasome constitutes a promising target for the treatment of various diseases, including cancer. Zhao et al. (2021) demonstrated the activation of the silent BGC for glidobactin-like NPs in the native host *Photorhabdus laumondii*, heterologous expression of this BGC in *Escherichia coli*, and crystal structure analysis, which could be a starting point to engineer this BGC for optimising the production of new bioactive compounds, providing a potential drug for cancer therapy [93].

In turn, Yang et al. (2018) reported that anthraquinones, such as 1,3,8-trihydroxy-9,10-anthraquinone and 3,8-dihydroxy-1-methoxy-9,10- anthraquinone, are promising NPs for the discovery of new neuroprotective and anti-neuroinflammatory drugs, preventing neurodegeneration of the central nervous system [94]. They effectively suppressed interferon-induced neuroinflammation of mouse microglial cells, decreasing the levels of nitric oxide, interleukin-6, and tumour necrosis factor-α. In addition, 1,3,8-trihydroxy-9,10- anthraquinone significantly protected mouse neuronal cells against cell death by inhibiting radical oxygen species (ROS) production, Ca^2+^ influx, and lipid peroxidation. 

### 3.2. Natural Products Derived from Xenorhabdus and Photorhabdus Bacteria with Antimicrobial Activities

The production of antimicrobial compounds by XB bacteria is necessary for the elimination of competitive microbes during the early and late stages of infections. Numerous antimicrobials acting against bacteria, fungi, and protozoa have been identified so far; however, for a long time, studies focused mostly on this effect toward plant pathogens [44,95] (Table 1 and Table 2). Antibiotic compounds derived from XP bacteria are structurally diverse peptides with most numerous nonribosomal peptides produced via 1/nonribosomal peptide synthetase (NRPS) (e.g., nematophin, odilorhabdin, PAX peptide, xenematide), 2/polyketides synthesised by polyketide synthetase (PKS) (e.g., anthraquinones, xenofuranone), and 3/NRPS-PKS hybrid compounds (e.g., cabanillasin, xenocoumacin). The others are ribosomal peptides (e.g., xenocin, xenorhabdicin, darobactin) or recently identified nucleoside triphosphates (ADG) [46,96,97]. Some antimicrobial metabolites of XP bacteria have been found to have potential for use as biocontrol agents, e.g., *Photorhabdus* stilbene derivatives showing antifungal activities against phytopathogenic fungi [98]; however, stilbenes also inhibited medically important fungi [42]. In addition, cabanillasin was found to be active against *Candida krusei* and *Candida lusitaniae* yeasts as well as the filamentous fungus *Fusarium oxysporum*, responsible for opportunistic infections in immunosuppressed patients [99]. 

Protozoa, such as *Acanthamoeba*, *Entamoeba*, *Trichomonas*, *Trypanosoma*, or *Leishmania* spp., are human parasites that cause chronic diseases and pose a threat to public health, especially in underdeveloped countries in Africa, Asia, and South America. Antiprotozoal activity has been exhibited by several compounds derived from XP bacteria, such as fabclavines, xenocoumacins, xenorhabdins, xenortides, rhabdopeptides, GameXPeptides, and PAX peptides [63,68,100]. Increasing resistance to artemisinin-based therapies in malaria-endemic regions makes the development of novel antimalarial drugs a compelling need. Chaiyaphumine A from *Xenorhabdus* spp. and rhabdopeptide/xenortide-like peptides from *X. innexi* have been shown to be promising compounds active against *Plasmodium falciparum*, the causative agent of malaria [63,101]. Furthermore, studies have shown that *P. luminescens* and *X. nematophila* can be sources of a new antitrypanosomal compound, a promising candidate as a new drug to fight Chagas disease [102]. It has been demonstrated that antitrypanosomal molecules secreted by both bacteria stimulate the trypanocidal activity of macrophages via a nitric oxide-independent mechanism. 

**Table 2 molecules-29-05151-t002:** List of specialised *Photorhabdus*-specific metabolites, their bioactivity, and biological functions.

BGC Product	Class/Biosynthetic Pathway *	Bioactivity	Biological Function/Mechanisms of Action	References
Anthraquinone	PKS	Antimicrobial, mosquitocidal, anti-neuroinflammatory, neuroprotective, probably ant and bird deterrent	Protection of neurons against induced cell death and suppression of neuroinflammation in mice	[84,94,103]
β-lactam carbapenem	Other	Antibacterial	Unknown	[104]
Carotenoid	Terpene	Antioxidant, cytotoxic, probably antibacterial	Unknown	[49,105]
Darobactin	RiPP	Antibacterial	Disrupting the outer membrane of Gram- bacteria by targeting BamA chaperone	[106]
Glidobactin/luminmycin	NRPS/PKS hybrid	Cytotoxic, antifungal	Proteasome inhibitor	[91,92,93]
Indigoidine	NRPS	Antioxidant, probably antibacterial	Probably protective function against ROS and UV	[105,107]
Isopropylstilbene/dialkylresorcinols/cyclohexanedione	Other	Antiprotozoal, antimicrobial, antioxidant, cytotoxic, immunomodulator	Quorum sensing, maintenance of nematode development—“food signal”	[108,109,110]
Kolossin	NRPS	Antiprotozoal	Unknown	[111,112]
Photopyrone	Other	Antibacterial	Quorum sensing	[113,114]
Photobactin	Siderophore	Antibacterial	Facilities the growth of bacteria in a Fe-limited environment, supports the growth and reproduction of nematodes	[115]
Ririwpeptide	NRPS	Unknown	Unknown	[49]

* NPRS, nonribosomal peptide synthetase; PKS, polyketide synthetase, RiPP—ribosomally synthesised and post-translationally modified peptides, ROS, radical oxygen species; UV, ultraviolet radiation.

## 4. Strategies for the Discovery of New Natural Products in *Xenorhabdus* and *Photorhabdus* Bacteria

For a long time, the NP research has been based on an extraction-based approach, including microbial cultivation, extraction of culture supernatants, and fractionation combined with activity testing. Advances in the metabolite profiling methods, especially NMR spectroscopy, liquid chromatography (LC), and high-resolution mass spectrometry (HMRS), facilitate the separation and identification of new compounds, making this approach more efficient. The strategy based on comparative metabolite analysis has yielded several new NPs involved in immunosuppression by inhibition of eicosanoid biosynthesis and insecticidal activity [116] and a new class of arginine-rich natural products in XP bacteria [117]. However, the extract-based approach is constrained due to the existence of numerous silent BGCs, which are not expressed in vitro or are expressed in low frequency, and compounds that are difficult to detect with standard analytical methods, e.g., membrane-bonds or compounds having untypical chemical properties. This suggests the need for genome mining, which has currently become the main driver in search of chemical novelty, usually being the starting point of workflow [57,118]. Genome-driven NP searches typically consist of three basic steps: BGC identification, structure prediction, and linkage of genomic to analytical methods, which are basically MS and NMR data derived from bacterial extracts [8]. Elements that have a tremendous impact on the efficient detection of new NPs are computational tools which are based on searching homologies using machine learning and molecular modelling, such as molecular networking, which helps reveal common features of identified BGC or chemical entities from sequence or analytical data, respectively [119]. This approach resulted in the identification of several new classes of NPs in XP bacteria, including xefoampeptides and tilivalline [120,121]. Genome mining may include pangenomic studies, which are based on the concept that genomes of bacteria belonging to the same prokaryotic species display large differences in the genetic information content, and only some genes are shared by all genomes [122]. The power of global analysis of BGCs combining pangenome and sequence similarity network methods was demonstrated by Shi et al. (2022) [49]. The analysis revealed that pangenomes of XP bacteria are open, i.e., the expected genetic variation is larger than observed (which is surprising as these bacteria are adapted to a very specific ecological niche), with a high frequency of gain of genes associated with secondary metabolism and loss of ancestral genes during evolution, explained by selecting valuable BGCs for efficient NP production as the driving force of their evolution.

### 4.1. Assessing Silent or Cryptic Biosynthetic Gene Clusters Through Genetic Engineering

Several strategies were employed to gain access to silent BGCs for further identification of their products, as well as the determination and elucidation of their activity and structure with the most commonly used genetic engineering methods [123]. Exchanging the native promotor of the BGC of interest with the arabinose-inducible promotor *P_BAD_* by homological recombination activated multiple BGCs in XB bacteria and led to the overexpression of desired NPs, e.g., GameXPeptides, xenoamicins, indigoidine [67], photoxenobactin, lipocitides [49], and siderophores [57], facilitated their isolation for bioactivity testing, or improved their yield for future application, as in the case of xenocoumacin 1 (Xcn1) [124]. Furthermore, the overriding of regulatory mechanisms by replacing the native BGC promoter with different constitutive promoters resulted, e.g., in improvement of the production of the blue pigment indigoidine in *P. luminescens* and fabclavines in *Xenorhabdus budapestensis* [107,125]. 

A strategy that has proven to be particularly effective in increasing the production of NPs of interest is combinatory engineering. The EasyPACId approach, based on the replacement of the natural promoter with the inducible promoter *P_BAD_* in RNA chaperone deletion mutants (ΔHfq), allows exclusive expression of desired NPs, enabling further analyses without time-consuming purification [126]. This approach yielded, e.g., the identification of new nematicidal NPs in *Xenorhabdus* bacteria, such as fabclavines, rhabdopeptides, and xenocoumacins [64]. In other research, an application strategy based on blocking degradation pathways, promotor exchange, and deletion of competing BGCs led to a large increase in Xcn1 production, helping commercial development [127]. These strategies improved the Xcn1 production efficiency more than others, involving only promotor exchange [124] or in situ product removal [128]. 

Polyketides (PKSs) and nonribosomal peptides (NRPS), the most widespread NPs in XP bacteria, are synthesised by huge multimodular enzymatic machineries, which build a molecular assembly line. Different re-engineering strategies have been used to restructure the existing domains and modules of PKSs and NRPSs during the last years in order to improve their yield, elicit their molecular structure, or produce novel compounds [129]. Re-engineering strategies consist of providing analogues of natural monomers, site-directed mutagenesis, or combinatorial biosynthesis. The generation of new rhabdopeptide/xenortide-like peptides by swapping short docking domains (ensuring the correct interaction of the NRPS subunits) and splitting the xefoampeptide BGC by introducing docking domains, yielding higher amounts of product, are examples of NRPS reengineering [130,131].

Since genetic manipulation in XP bacteria is more challenging than in model bacteria, such as *E. coli*, to overcome this limitation, Yin et al. (2015) implemented a lambda Red recombineering system to facilitate promotor insertion leading to activation of the NRPS gene cluster in *P. luminescens* [111]. Later, specific for *Xenorhabdus* spp., the RecET-like recombineering system was developed and used for the removal of BGC of interest, which led to the identification of a new cyclopeptide changshamycin, which exhibited antibiotic and cytotoxic properties [132].

When XP strains are insufficiently amenable to genetic manipulation in natural hosts, heterological expression may be a feasible alternative; however, challenges can be posed by improper biosynthesis or toxicity of the overproduced product. Heterologous expression in the most popular host *E. coli* resulted in the identification of many new NPs from XP bacteria, such as indigoidine [107,121], ambactin, xenolindicin [133], or methionine-containing rhabdopeptide/xenortide-like peptides [63]. Heterologous expression in *E. coli* was also helpful for establishing the natural role of a given NP, for example, rhabduscin as a potent nanomolar-level inhibitor of phenoloxidase [65] or the identification of biosynthetic steps, as in the case of the antibacterial 3′-amino 3′-deoxyguanosine (ADG) prodrug [97]. Other heterological hosts have also been reported, e.g., *Pseudomonas putida* for the expression of syrbactin synthetase [134] or closely related *X. szentirmaii* for the heterologous overexpression of *X. nematophila leuO*, which resulted in the overproduction of several novel bioactive compounds, including ubiquitous GameXPeptides [30].

### 4.2. Concentration of Bacterial Extracts, Alteration of Growth Condition, and Chemical Synthesis

A non-standard approach, which allows high-throughput NP research, is the screening of highly concentrated cell extracts. This approach yielded darobactin from *Photorhabdus khanii* [106], while screening concentrated *P. luminescens* extracts up to 200× resulted in the identification of new antibiotic compound ADG, a mimic of GTP [97]. In addition, many studies demonstrated that the production of secondary metabolites could be improved by altering the XP growth conditions, especially manipulating the pH value, temperature, medium volume, rotary speed, inoculation volume, or medium composition [46,96]. For example, optimisation of the environmental parameters in *X. nematophila* using the response surface methodology resulted in 43% higher antibiotic activity than in non-optimised conditions [135]. More recently, Booysen et al. (2021) have reported that various secondary metabolites are produced in *Xenorhabdus khoisanae* in different conditions of oxygen and water supply [136]. The modification of the medium composition by selecting appropriate carbon, nitrogen, and mineral sources increased the antifungal activity of *X. stockiae* up to 27%, compared to standard medium [137]. It was also observed that XP bacteria produced more secondary metabolites when cultured on insect’s hemolymph. Further metabolomic profiling revealed that the metabolic shift initiates one of its compounds, L-proline, affecting virulence and antibiotic production [138]. In the case of isopropylstilbene (IPS), nutrient limitation and the presence of aromatic amino acids were shown to be crucial for the regulation of the expression of the *stlA* gene, which encodes the enzyme catalysing the first step of IPS biosynthesis [139]. In turn, the production of hydroxymate siderophores in XP needs iron starvation conditions [57].

For a few NPs with a known and relatively simple synthetic pathway, chemical synthesis may be suitable for further investigation and application, especially in the case of low productivity. Bactericidal darobacin is a good example, as this substance, together with more active analogues, has recently been obtained via chemical synthesis by independent researcher groups [106]. Chemical modification of naturally produced precursors could also be a good resolution, as in the case of fabclavines with a broad range of activity; moiety substitution may help in obtaining changed, more targeted activity [140].

### 4.3. Biosynthetic Gene Cluster Analysis in Xenorhabdus and Photorhabdus Bacteria

Genome mining has revealed that XP bacteria dedicate a significant proportion of their genome (6–7.5%) to the production of NPs [121]. The number of BGCs detected per genome in XP bacteria far exceeds those expected based on the NPs discovered using the extract approach, ranging usually from 20 to 40 BGCs per strain; however, as many as 65 BGCs per strain have also been reported [49,120,141]. Notably, this highly exceeds the BGC number in the genomes of closely related *Enterobacteriaceae* [142]. There is a correlation between the genome size and the number of BGCs, thus *Photorhabdus* representatives with a larger genome contain on average more BGCs than *Xenorhabdus.* On the other hand, *Xenorhabdus* bacteria have on average more genus-specific and strain-specific BGCs [49]. Extensive studies have demonstrated an impressive diversity of BGC classes and their relative metabolites in XP bacteria [120,141]. As estimated, more than half of the total BGCs encoded in XP genomes, which represent a high range of molecular structures, is unique for these bacteria; thus, they are particularly valuable for NP research. From these BGCs, XP bacteria are anticipated to produce a huge structural variation employing different mechanisms, e.g., module skipping, precursor promiscuity, or alternative transcriptional starting [66,143,144]. For example, the chemical diversification of rhabdopeptide/xenortide-like peptides results from a combination of iterative and flexible use of monomodular NRPS, including substrate promiscuity, enzyme cross-talk, and enzyme stoichiometry, as shown by experiments conducted by Cai et al. (2017) [62]. The most abundant NPs in XP are nonribosomal peptides synthesised by NRPS, which account for over half of all BGCs, suggesting their essential ecological functions. The others belong to PKS/NRPS hybrid compounds, polyketides being a product of PKS, and relatively few ribosomal synthesised and post-translationally modified peptides (RiPP), terpenes, and various minor NP classes [49].

Research has demonstrated that, despite life cycle similarities and phylogenetic proximity, XP bacteria show significant genomic divergence [145]. This is reflected in the fact that the set of identified BGCs and discovered metabolites in these two genera differ and only a few BGCs are conserved, pointing to differences between XP bacteria in their interactions with hosts. BGCs responsible for the synthesis of GameXPeptides defeating insect immune response, insecticidal photoxenobactin, and the proteasome inhibitor β-lactone as well as antiprotozoal rhabdopeptide/xenortide-like peptides and the immune modulator rhabduscin are the most conserved BGCs present in these two bacterial genera [49,120]. The *Xenorhabdus*-specific BGCs comprise those coding for multiactive fabclavine, antiprotozoal xenoamicin, and aryl polyene lipids, which are protective factors against oxidative stress and contribute to biofilm formation in *E. coli* [146]. In turn, the most prevalent *Photorhabdus*-specific BGCs are those for multipotent isopropylstilbene, ririwpeptide with unknown function, and cytotoxic glidobactin (Table 1, Table 2 and Table 3). In XP bacteria, strain-level variations in the NP spectrum have also been frequently reported [147]. For example, althiomycin, andrimid, and malonomycin BGCs were detected separately in specific strains of *Xenorhabdus indica*, *Photorhabdus temperata*, and *Photorhabdus akhurstii*, respectively [118]. Regarding siderophores, analyses have revealed that putrebactin- and avaroferrin-producing BGCs are more widespread and were most likely present in a common ancestor of these bacteria; however, the aerobactin and ochrobactin BGCs were probably taken up by only a few strains individually [57]. 

Widespread NP families susceptible to lateral gene transfer are usually metabolically undemanding and related to general physiological advantages. For example, glidobactin A (syrbactin family), which acts as a proteasome inhibitor, apart from *Photorhabdus* bacteria, were identified in many other pathogenic bacteria. Other promiscuous NPs identified in XP were the blue dye indigoidine, attributed to the protective role of oxygen radicals [107], and tilivalline, produced also by *Klebsiella oxytoca* [121], which belongs to the class of NPs known for their cytotoxicity. Presented in all genomes of studied XP, the *ioc/leu* BGC responsible for the production of proteasome inhibiting β-lactone is also widely distributed in other γ-proteobacteria. Studies focusing on the discovery of NPs showing high chemical similarity to already known metabolites are noteworthy, as they are likely to encode novel analogues, being also a source of great novelty.

## 5. *Xenorhabdus* and *Photorhabdus* Metabolites as Promising Drug Leads

Many NPs produced by XP bacteria do not meet the needs of the pharmaceutical sector because they are too multifunctional, display undesirable cytotoxic properties, or have a high molecular weight. However, some selected NPs are promising for development into new therapeutics, given their tumour-targeting activity, good pharmacokinetics, or attractive mode of action. Especially difficult to find are antibiotics targeting Gram- bacteria due to the highly restrictive permeability of their outer membrane. However, XP bacteria share, in fact, similar needs for antibiotic protection as humans to fend off Gram- competitors in the insect larvae that they infect. Notably, bioactive compounds produced by XP bacteria must be nontoxic to their nematode host and have good exposure in insect hemolymphs, predisposing these bacteria to be a source of new drugs.

### 5.1. Darobactin

Extensive screening of concentrated extracts from *Photorhabdus* strains resulted in the discovery of a new antibiotic produced by *P. khanii*, named darobactins [106]. It is a relatively large molecule with a unique scaffold of modified heptapeptide containing two unusual crosslinks: the C-O-C Trp-Trp ether bond and the C-C Trp-Lys bond (Figure 3). This metabolite belonging to the RiPP class is encoded by a silent operon consisting of *darA*, *darBCD*, and *darE* ORFs encoding, respectively, propeptide, ABC-type envelope exporter, and highly versatile radical S-adenosylmethionine (SAM), which is responsible for forming a unique bicyclic structure [165]. Genome mining studies have shown that *dar* operons are common in *Photorhabdus* and have also been identified in several other bacteria associated with animals, such as *Yersinia*, *Vibrio*, and *Pseudoalteromonas*. The significantly lower GC content of the *dar* operon, compared to these of *Photorhabdus khanii* and other proteobacteria, suggests its horizontal acquisition; however, the donor is not known. Darobactin targets BamA, the essential β-barrel chaperone of the outer membrane of Gram- bacteria, by stabilising its gate-closed conformation, resulting in the disruption of outer membrane function [106,166]. In fact, targeting the chaperone, not the catalytic centre of the enzyme, is a relatively unusual mode of action. This antibiotic is active against many Gram- pathogenic bacteria, including multi-drug resistant (MDR) strains (e.g., *E. coli* MIC = 4 μg/mL), but not Gram- human gut symbionts. Darobactin has also shown good efficiency in mouse septicaemia and a mouse thigh infection with *E. coli*. The heterologous expression of a synthetically engineered darobactin BGC in *E. coli* and bioengineering studies contributed to good yields and improved the spectrum of activity of this compound [167,168]. Recently, the chemical synthesis of this promising drug lead has also been achieved [165].

### 5.2. Dynobactin A

Another class of antibiotics active against Gram- pathogens are dynobacins. They were characterised using computational analysis of BGCs distantly related to those of darobactin, which found a large DynA clade encoding RiPP peptides targeting BamA, i.e., the same chaperone as darobactins [169]. Structural analyses based on cryo-electron microscopy and micro-electron diffraction revealed that dynobactin A from *Photorhabdus australis* is a 10-mer peptide with two unique cyclophane rings. In particular, it is not toxic to mammalian cells up to 1000 µg ml^−1^, has good solubility in water (excess 200 mg mL^−1^), and displays strong target binding strength. Compared to darobactin, dynobactin A shows higher in vitro potency but is 4-fold less efficient against clinically relevant *E. coli* pathogens (*E. coli* MIC = 16 μg/mL), suggesting penetration through the outer membrane as a limiting factor. Nevertheless, the favourable properties and successful total synthesis of dynobactin A [170] ensure wide access to this type of NP and its analogues, providing a great opportunity for future antibiotic development [171].

### 5.3. Odilorhabdins (ODLs)

Cationic linear peptide ODLs are a newly described class of NPs from *Xenorhabdus.* ODLs are produced in *X. nematophila* by NRPS gene clusters. By binding to the decoding part of the small ribosomal subunit, they inhibit protein synthesis, making it highly prone to translation errors [52]. Importantly, the binding sites of other known ribosome miscoding antibiotics, e.g., the relatively toxic aminoglucosides, are located differently; hence, mutation of the ribosome decoding centre that would affect their binding and action is not expected to disturb ODL activity. One of the ODLs, named NOSO-95179 (Figure 4), effectively acts against a wide spectrum of pathogenic Gram+ and Gram- bacteria, including carbapenem-resistant *Enterobacteriaceae* (*E. coli* MIC = 8–16 μg/mL, *Klebsiella pneumoniae* MIC = 4–8 μg/mL) or methicillin-resistant *Staphylococcus aureus* (MIC = 16 μg/mL) (Figure 5). In particular, it does not show cytotoxicity to human HepG2 and HK-2 cells even at a concentration of 256 μg/mL. In addition, it exhibits safety and promising therapeutic efficiency in mouse models of *K. pneumonia* septicemia and lung infection [52]. Experimental efforts using de novo chemical synthesis led to the development of several ODL derivatives with optimised pharmacological properties, and one of them, NOSO-502, was selected as the best preclinical candidate [172]. NOSO-502 shows very high efficacy against *Enterobacteriaceae*, including strains with a carbapenem-resistant phenotype, with MIC values ranging from 0.5 to 4 μg/mL. Its efficacy in several clinically relevant animal infection models and its favourable in vitro safety profile were demonstrated as well [53,54,173]. This raises hopes that ODLs could be clinically useful antibiotics without toxicity and a low frequency of pathogen resistance. The mechanism of self-protection against ODLs involving *N*-acetyltransferase present in host cells has already been described [174].

### 5.4. Isopropylstilbene (IPS)

As an important class of bioactive metabolites, stilbenes are generally associated with plants. The best-known plant stilbene is resveratrol, with numerous proven health advantages for humans. IPS, i.e., 3,5-dihydroxy-4-isopropylstilbene, has gained researchers’ attention mainly due to its multipotent biological properties, including antimicrobial, antiprotozoal, antioxidant, and immunosuppressive activities. BGCs for IPS were found in the genomes of all the *Photorhabdus* strains studied so far. IPS acts as a signal for the nematode that stimulates the recovery of IJ to the adult hermaphrodite, allowing coordination of nematode development with bacterial growth, while its antimicrobial properties help eliminate a wide range of competitors [110]. Analyses have revealed the most intensive IPS expression in the post-exponential phase of growth when cultured in vivo, while its level after *G. mellonella* infection increases from 24 h post infection and remains stable for several days. Multiple steps of regulation that affect the level of IPS in *Photorhabdus* cells have also been described [139,175,176]. The branched biochemical pathway of IPS production in *Photorhabdus*, involving both the metabolism of fatty acids and amino acids, was elucidated by Joyce et al. (2008) [108]. It is distinct in several aspects from the plant counterpart; therefore, it represents an essential source of chemical novelty. The mechanism of stilbene cell detoxification and its exporting by the AcrAB efflux pump in *Photorhabdus* spp., contributing to the ability of these bacteria to survive under high concentrations of stilbenes, which are toxic in high amounts to bacteria, was described earlier [177,178].

In 2011, the evaluation of an oil-in-water cream containing 3,5-dihydroxy-4-isopropylstilbene for the topical treatment of plaque psoriasis was undertaken in clinical studies in China and Canada. The promising results resulted in the development of phase 3 trials, which showed that the cream is effective and well tolerated by patients with mild to severe plaque psoriasis and atopic dermatitis [179,180]. Finally, a 1% cream was approved and is currently sold in China and the United States as an anti-inflammatory and antimicrobial therapeutic agent under the names benvitimod and tapinarof, respectively [181]. It is regarded a breakthrough topical treatment for plaque psoriasis since vitamin D analogues were marketed in the 1980s; however, the mechanism of IPS action is not yet fully understood [182].

## 6. Summary

As species-specific mutualists of insect-infecting entomopathogenic nematodes, *Xenorhabdus* and *Photorhabdus* bacteria deserve attention as a goldmine for novel therapeutics, especially including antibiotics, anticancer, and immunosuppressive drugs. Two novel antibiotics, i.e., ribosomal inhibitors odilorhabdins and outer membrane-perturbing darobactin, are good examples of success in finding novel compounds acting against Gram- bacteria, including MDR strains, which are a critical priority for human health. Notably, the metabolites identified so far are only a small part of new chemistry harbouring by these bacteria, which are easy to cultivate in the laboratory and amenable to molecular manipulation. Thus, there are still many more NPs to be discovered, especially considering that a number of new strategies have been developed to address the limitation of NP research. For example, re-engineering of modular NRPS and PKS systems, facilitated by such novel approaches as CRISPR/Cas or synthetic zippers, seems to lead to very promising results with regard to the production of novel NPs, improvement of their activity or yield, and acquisition of analogues of known compounds, despite the obstacles and high failure rate. The development in genome analysis has a tremendous impact on NP discovery; however, the main bottlenecks are the activation of interesting BGCs, linking the BGC with the metabolite, and prioritising BGCs for further research. Because of the limitation of bioinformatics tools in the prediction of metabolite scaffolds, the biosynthetic potential of XP should be validated experimentally. 

Despite the success in identifying new compounds with possible medical applications, the most difficult issue is to elucidate their biological role due to the complexity of natural communities; nevertheless, these are still challenging studies despite the possibility of in-depth investigations using the ‘omics’ technology. In the case of NPs with antibiotic activity, it seems quite clear that they are responsible for defence in the natural environment; however, these substances in low concentrations may play a role in signalling rather through regulation of the level of gene expression in the cell. Further analyses of their targets in signalling pathways and synergistic interactions can help to decipher in what way the NP interplay controls the symbiotic and pathogenic lifestyle and, as a result, accelerates drug discovery. Studying XP bacteria could be a starting point to understanding other more complex symbionts, especially the human microbiota. Another interesting issue can be to find out in what way cognitive nematodes and symbionts survive in an insect cadaver, which is full of toxic compounds. Regarding the mechanisms of resistance against toxic NPs, an uncommon prodrug strategy, which protects bacteria from the toxicity of antimicrobial metabolites by assembling a non-toxic precursor, could enhance the bioavailability and potency of virulence factors, as in the case of AGD, an antibacterial agent discovered recently.

## Figures and Tables

**Figure 1 molecules-29-05151-f001:**
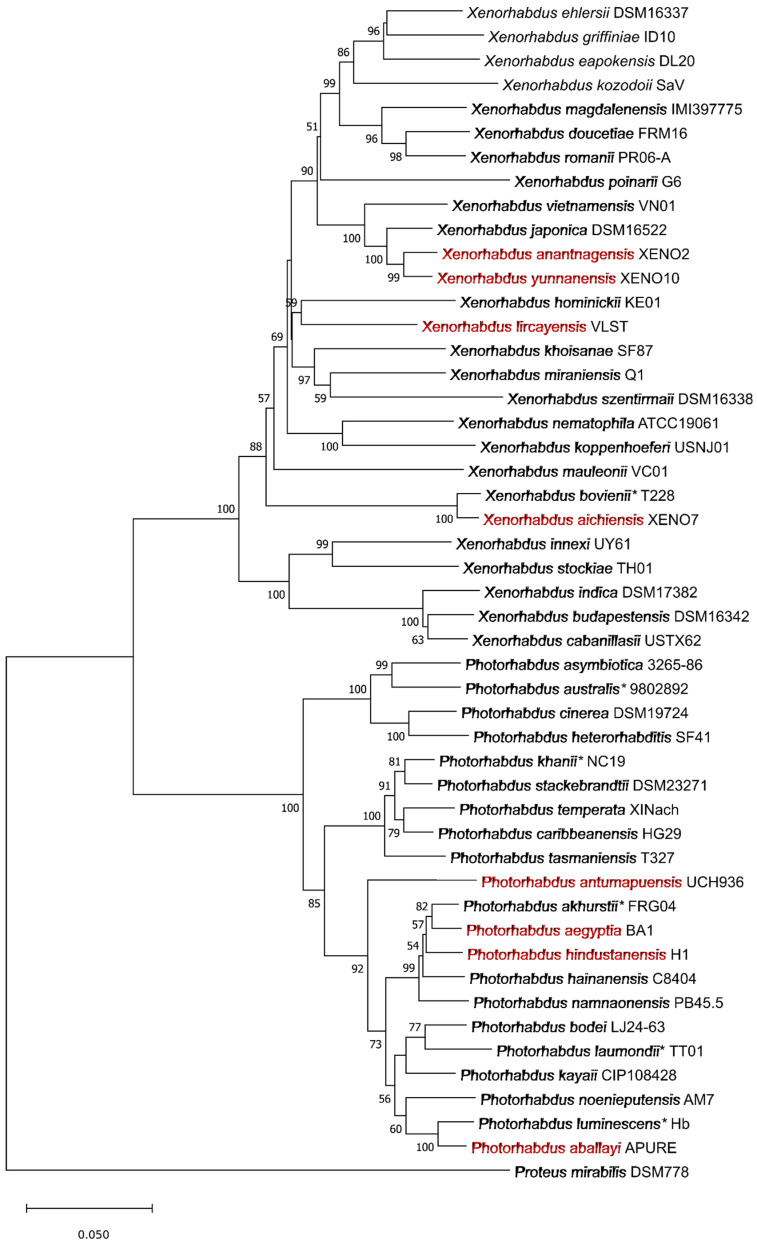
Neighbour-joining phylogenetic tree of the type strains of the *Xenorhabdus* and *Photorhabdus* species reconstructed from the concatenated nucleotide sequences of *recA*, *dnaN*, *gltX*, and *gyrB* genes (2816 bp). The sequences were retrieved from the GenBank using BLAST (NCBI). Multiple sequence alignment was created using ClustalW and the evolutionary distances were computed using Maximum Composite Likelihood method in MEGA11. Bootstrap values [%] greater than 50 are shown next to the branches. The scale bar represents 0.05 substitutions per nucleotide position. The new bacterial species of the *Xenorhabdus* and *Photorhabdus* genera (identified from 2021 onwards) are shown in coloured font, while species that include subspecies are marked with an asterisk.

**Figure 2 molecules-29-05151-f002:**
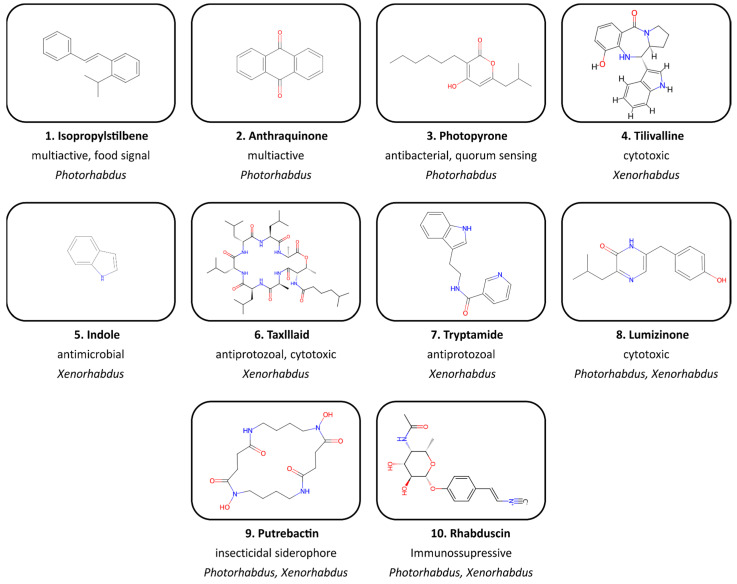
Examples of chemical structures of bioactive compounds discovered in *Xenorhabdus* and *Photorhabdus* bacteria.

**Figure 3 molecules-29-05151-f003:**
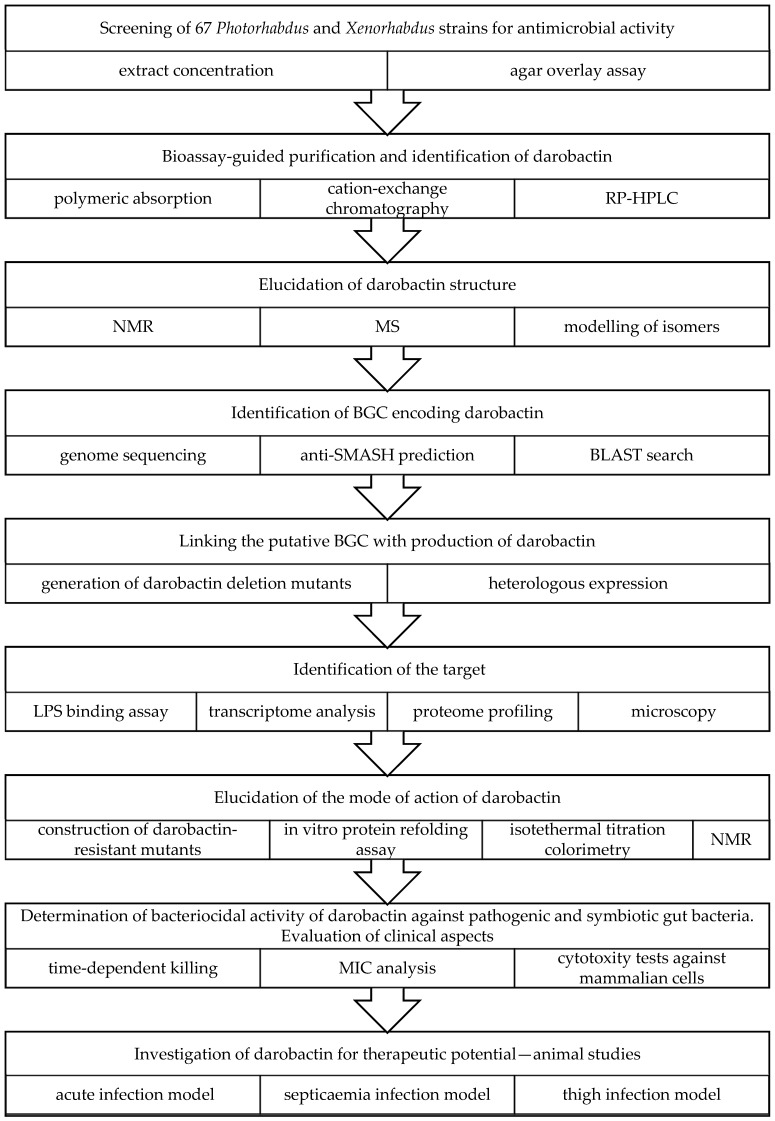
Darobactin discovery scheme [106].

**Figure 4 molecules-29-05151-f004:**
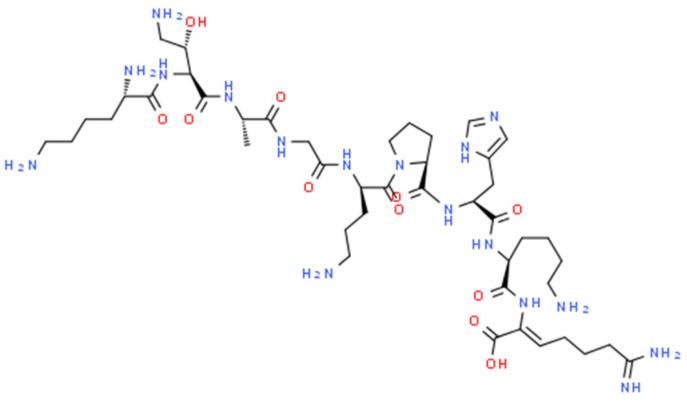
Chemical structure of NOSO-95179.

**Figure 5 molecules-29-05151-f005:**
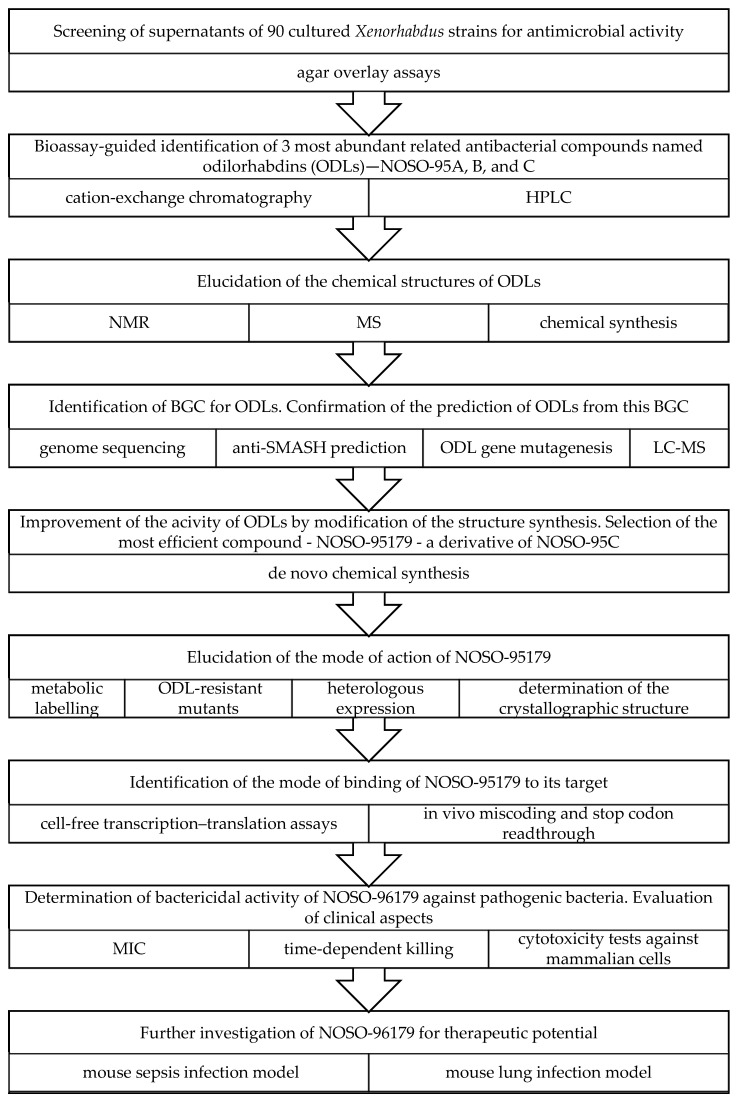
Discovery scheme of odilorhabdins [52].

**Table 3 molecules-29-05151-t003:** List of specialised *Xenorhabdus*-specific metabolites, their bioactivity, and biological functions.

BGC Product	Class/Biosynthetic Pathway *	Activity	Biological Function/Mechanism of Action	References
Ambactin	NRPS	Antiprotozoal	Unknown	[133]
Aerobactin	Siderophore	Insecticidal	Virulence-related metalloprotein	[57]
Aryl-polyene lipid	Other	Unknown	Biofilm formation, protection against oxidative stress	[49,146]
Benzylideneacetone	Other	Antibacterial, immunosuppresive	Inhibition of phospholipase A2, inhibition of haemocyte nodule formation	[75,148]
Bicornitun	NRPS	Antimicrobial cytotoxic	Unknown	[117]
Cabanillasin	NRPS/PKS hybrid	Antifungal	Unknown	[99]
Fabclavine	NRPS/PKS/ PUFA hybrid	Antibacterial, antiprotozoal, antifungal, nematocidal, insecticidal, cytotoxic	Disruption of midgut epithelial cells and pH balance	[68,81,82,149]
Indole/oxindole	Indole	Antifungal, antibacterial, insecticidal, immunosuppressive, cytotoxic, nematocidal	Phospholipase A2 inhibitor	[75,86,116]
Lipocitide	NRPS	Unknown	Inhibition of the nitric oxide pathway	[49]
Nematophin	NRPS	Antibacterial	Unknown	[150]
PAX peptide	NRPS	Antibacterial, antifungal, antiprotozoal	Unknown	[51,68,151,152]
Szentiamide	NRPS	Antiparasitic, cytotoxic	Disruption of haemocytes	[153,154]
Taxlllaid	NRPS	Antiprotozoal, cytotoxic	Unknown	[155]
Tilivalline	NRPS	Cytotoxic	Disruption of gut epithelial cells	[121,156]
Rhabdopeptides	NRPS	Cytotoxic, antiprotozoal, hemotoxic	Unknown	[60]
Xefoampeptide	NPRS	Insecticidal	Unknown	[120]
Xenobactin	NRPS	Antiparasitic, antibacterial		[157]
Xenoamicin	NRPS	Antiprotozoal, cytotoxic	Probable interaction with cytoplasmic membrane	[158]
Xenocoumacin	NRPS/PKS hybrid	Antibacterial, antifungal, nematocidal, insecticidal, antiprotozoal, acaricidal, anti-ulcer	Inhibition of mRNA translation. Prodrug activation mechanism	[64,68,85,159,160]
Xenocycloin	PKS	Insecticidal, cytotoxic	Disruption haemocytes	[161]
Xenofuranone	PKS	Cytotoxic, antibacterial	Unknown	[162]
Xenotetrapeptide	NRPS	Insecticidal, cytotoxic	Unknown	[163]
Xentrivalpeptide	NRPS	Unknown	Unknown	[164]
Xenortide/ tryptamide	NRPS	Antiprotozoal, cytotoxic	Unknown	[61,87]

* NPRS, nonribosomal peptide synthetase; PKS, polyketide synthetase.

## Data Availability

Not applicable.

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
