# Peer review of "Exploring Xenorhabdus and Photorhabdus Nematode Symbionts in Search of Novel Therapeutics"

_molecules, 2024, doi:10.3390/molecules29215151_

Round 1

Reviewer 1 Report

Comments and Suggestions for Authors

This article is a review of the literature on the therapeutic potential of molecules produced by the symbiotic bacteria, Xenorhabdus and Photorhabdus (referred as XP). The article focuses on natural products, which are mostly molecules produced by NRPS, PKS, and NRPS/PKS hybrids, along with several ribosomal peptides. The authors provide a comprehensive inventory of natural products in XP, accompanied by summary tables. They discuss active natural products against animal cells, antimicrobial natural products, and cytotoxic natural products. They list new promising strategies for the discovery of novel molecules. Finally, the authors describe in more detail some examples of molecules of therapeutic interest that are in advanced development as promising drug leads.

Major concerns

- Chapters 3.1 and 3.3 should be merged, as both sections address natural products targeting animal (eukaryotic) cells.

- Chapter 5.5 is not relevant to the topics of the review and should be removed. While insecticidal toxins are promising for agriculture as insecticides, their application as therapeutic products in human and animal health is unclear. Similarly, PVCs are not considered natural products.

Minor concerns

- Line 77 : « Importantly, natural  competitors during the life cycle of XP bacteria are largely Gram- proteobacteria, which gives hope of discovering antibiotics targeting Gram- pathogens, particularly needed today » :

No, the competitors originate from various environmental sources or from the insects themselves and include both Gram-positive and Gram-negative bacteria, as well as fungi, among others.

- Line 105 to 114 : references are missing to support this section

- Line 113: The statement 'sterilization of the cadaver' is inaccurate. In fact, the cadaver is not sterile; it functions as a biofermentor containing numerous microorganisms.

- Line 142: The statement 'but also as plant growth-promoting bacteria' is inaccurate. While Xenorhabdus and Photorhabdus exhibit antifungal activities against plant pathogenic fungi, they do not demonstrate plant growth-promoting rhizobacteria (PGPR) activities.

- The statement about « Other compounds with antimicrobial activity are polyketides (e.g. anthraquinones, xenofuranone)» is misleading. The authors differentiate between peptides synthesized from NRPS/PKS (lines 269 to 274) and polyketides (lines 274 to 275), while polyketides are assembled via PKS enzymes

- Lane 584 : Gramm – ?

- Line 610 : « Relatively simple in structure », Not really simple, as the amino acids in position 2 and 3 (beta-hydroxy Dab) or in position 9 (dehydroarginine) are extremely rare in nature and essential for the activity.

- Line 615 : Replace « aminoglucosidases » with aminoglycosides

- Line 620 : The MIC values are not correct : E. coli MIC = 8 to 16 μg/ml, Klebsiella pneumoniae MIC=4 to 8 μg/ml

- Lines 622 to 624 : There is significant confusion in this section. The authors are supposed to describe NOSO-95179, but instead mixes results obtained with NOSO-95179 and, more importantly, NOSO-502 (ref 49). Moreover, in the cited article (48), the tested cells are HepG2 and HK-2, and the threshold value is 256 μg/ml

- Line 708 : The metabolites identified so far are only a smart part :  small part ?

- Fig. 3 (Darobactin): construction of odilobactin resistant mutants ?

References :

There are numerous errors in the references.

- Many italics are missing, such as for genus and species names or gene names.

- Ref. 64 : The DOI does not correspond to the cited reference

- Ref. 79 is missing the publication year

- Ref. 40 is missing the DOI

- Ref. 115 is missing the DOI, which should be doi: 10.1007/82_2016_24

Certain review articles are frequently cited, such as Ref. 43; the authors should reference the original publications.

Author Response

We would like to thank the Reviewer for the useful comments to improve the paper. We have addressed all the comments as explained below.

Comment 1: “Chapters 3.1 and 3.3 should be merged, as both sections address natural products targeting animal (eukaryotic) cells”

Replay: According to the Reviewer’s remark, both sections have been merged into one (3.1)  under the title: “Natural products derived from Xenorhabdus and Photorhabdus bacteria targeting eukaryotic cells”

Comment 2: “Chapter 5.5 is not relevant to the topics of the review and should be removed. While insecticidal toxins are promising for agriculture as insecticides, their application as therapeutic products in human and animal health is unclear. Similarly, PVCs are not considered natural products.”

Replay: It has been corrected. The chapter 5.5 has been removed from the manuscript.

Comment 3: “Line 77”,  “Importantly, natural  competitors during the life cycle of XP bacteria are largely Gram- proteobacteria” “No, the competitors originate from various environmental sources or from the insects themselves and include both Gram-positive and Gram-negative bacteria, as well as fungi, among others.”

Replay: It has been corrected. This sentence has been removed from the manuscript.

Comment 4: “Line 105 to 114 : references are missing to support this section”

Replay: It has been corrected.  Following references have been added to the aforement5ioned part of the manuscript:

  1. Koppenhöfer, H.; Gaugler, R. Entomopathogenic Nematode and Bacteria Mutualism. In Defensive Mutualism in Microbial Symbiosis; White, J., Torres, M., Eds.; CRC Press, 2009; Vol. 20090677 ISBN 978-1-4200-6931-0 978-1-4200-6932-7.
  2. Herbert, E.E.; Goodrich-Blair, H. Friend and Foe: The Two Faces of Xenorhabdus nematophila. Nat Rev Microbiol 2007, 5, 634–646, doi:10.1038/nrmicro1706.
  3. Clarke, D.J. Photorhabdus : A Model for the Analysis of Pathogenicity and Mutualism. Cellular Microbiology 2008, 10, 2159–2167, doi:10.1111/j.1462-5822.2008.01209.x.

Comment 5: “Line 113: The statement 'sterilization of the cadaver' is inaccurate. In fact, the cadaver is not sterile; it functions as a biofermentor containing numerous microorganisms.”

Replay: It has  been corrected. The phrase 'sterilization of the cadaver'has been removed. Additionally, the entire sentence has been transformed so that the processes occurring in the first phase of infection are better covered: ‘Within the nutritional insect cadaver, the bacteria continue to grow and produce metabolites to ensure degradation of insect tissue, development of nematode progeny, suppression of microbial competitors, and deterrence of opposing nematodes and saprophytic scavengers.’

Comment 6: “Line 142: The statement 'but also as plant growth-promoting bacteria' is inaccurate.”

Replay: It has  been corrected . Aforementioned  statement has been removed. A more general statement was left: “Therefore, further studies of XP bacteria may open the door to future applications of these bacteria not only as biopesticides in agriculture but also as  plant protecting agents”.

Comment 7: “The statement about “Other compounds with antimicrobial activity are polyketides (e.g. anthraquinones, xenofuranone)” is misleading. The authors differentiate between peptides synthesized from NRPS/PKS (lines 269 to 274) and polyketides (lines 274 to 275), while polyketides are assembled via PKS enzymes”.”

Replay: These statements have been corrected to be more clear: “Antibiotic compounds derived from XP bacteria are structurally diverse peptides with most numerous nonribosomal peptides produced via 1/ nonribosomal peptide synthetase (NRPS) (e.g., nematophin, odilorhabdin, PAX peptide, xenematide), 2/polyketides synthesised by polyketide synthetase (PKS) (e.g. anthraquinones, xenofuranone),  and 3/ NRPS-PKS hybrid compounds (e.g., cabanillasin, xenocoumacin). The others are  ribosomal peptides (e.g., xenocin, xenorhabdicin, darobactin) or recently identified nucleoside triphosphates (ADG) [46,96,97].”

Comment 8: “Lane 584 : Gramm – ?”

Replay: According to the reviewer’s remark, this part of the sentence was changed to more precise: “This antibiotic is active against many Gram- pathogenic bacteria.”

Comment 9: “Line 610 : “ Relatively simple in structure”, “Not really simple, as the amino acids in position 2 and 3 (beta-hydroxy Dab) or in position 9 (dehydroarginine) are extremely rare in nature and essential for the activity.”

Replay: It has been corrected. The statement ‘Relatively simple in structure” has been removed from the sentence. Now that sentence reads as follows: “Cationic linear peptide ODLs are a newly described class of NPs from Xenorhabdus. ODLs are produced in X. nematophila by NRPS gene clusters.”

Comment 9: “Line 615 : Replace « aminoglucosidases » with aminoglycosides”

Replay: It has been corrected.

Comment 10: “Line 620 : The MIC values are not correct : E. coli MIC = 8 to 16 μg/ml, Klebsiella pneumoniae MIC=4 to 8 μg/ml”

Replay: It has been corrected.

Comment 11: “ Lines 622 to 624 : There is significant confusion in this section. The authors are supposed to describe NOSO-95179, but instead mixes results obtained with NOSO-95179 and, more importantly, NOSO-502 (ref 49). Moreover, in the cited article (48), the tested cells are HepG2 and HK-2, and the threshold value is 256 μg/ml”

Replay: It has been corrected strictly according to the reviewer’s remark. The description of NOSO-95179 and NOS-502 was separated. A more detailed description of NOSO-502 has been supported by references from the relevant literature [54, 174]. Data on MIC values and cell assays have been updated. Currently, this part is as follows: “One of the ODLs, named NOSO-95179, effectively acts against a wide spectrum of pathogenic Gram+ and Gram- bacteria, including carbapenem-resistant Enterobacteriaceae (E. coli MIC = 8-16 μg/ml, Klebsiella pneumoniae MIC=4-8 μg/ml) or methicillin–resistant Staphylococcus aureus (MIC=16 μg/ml) (Fig. 4). In particular, it does not show cytotoxicity to human HepG2 and HK-2 cells even at a concentration of 256 μg/ml. In addition, it exhibits safety and promising therapeutic efficiency in mouse models of K. pneumonia septicemia and lung infection [52]. Experimental efforts using de novo chemical synthesis led to the development of several ODL derivatives with optimised pharmacological properties, and one of them, NOSO-502, was selected as the best preclinical candidate [172].  NOSO-502 shows very high efficacy against Enterobacteriaceae, including strains with a carbapenem-resistant phenotype, with MIC values ranging from 0.5 to 4 μg/ml. Its efficacy in several clinically relevant animal infection  models and its favourable in vitro safety profile were   demonstrated as well  [53,54,173].

Comment 12: “Line 708 : The metabolites identified so far are only a smart part :  small part ?”

Replay: It has been corrected.

Comment 13: “Fig. 3 (Darobactin): construction of odilobactin resistant mutants ?”

Replay: It has been corrected.

Comment 14: “Many italics are missing, such as for genus and species names or gene names.”

Replay: It has been corrected.

Comment 15: ‘Ref. 64 : The DOI does not correspond to the reference cited. ‘

Replay: This DOI number  has been removed. In online libraries, just such a DOI number was assigned to the aforementioned article, hence the mistake.  However, it does not have the right connection to the article, which is why it was finally removed from the Literature section.

Comment 16: “Ref. 79 is missing the publication year”

Replay: The publication year (2012) has been bolted to make it more visible.

Comment 17: “Ref. 40 is missing the DOI”

Replay: This article is not assigned a DOI number, as is the current ref. 24 and 41.

Comment 18: “Ref. 64 : “Ref. 115 is missing the DOI, which should be doi: 10.1007/82_2016_24”

Replay: It has been corrected.

Comment 19: “Certain review article are frequently cited, such us Ref. 43, the authors should reference the original publications”

Replay: It has been corrected. In Line 221 Ref. 43 was replaced with the 2 original articles:

  1. Sanda, N.B.; Hou, Y. The Symbiotic Bacteria—Xenorhabdus nematophila All and Photorhabdus luminescens H06 Strongly Affected the Phenoloxidase Activation of Nipa Palm Hispid, Octodonta nipae (Coleoptera: Chrysomelidae) Larvae. Pathogens 2023, 12, 506, doi:10.3390/pathogens12040506.
  2. Kenney, E.; Hawdon, J.M.; O’Halloran, D.; Eleftherianos, I. Heterorhabditis bacteriophora Excreted-Secreted Products Enable Infection by Photorhabdus luminescens Through Suppression of the Imd Pathway. Front. Immunol. 2019, 10, 2372, doi:10.3389/fimmu.2019.02372.

In addition, the recurring item 21 (Sajnaga et al. 2020) was removed and replaced (Line 116):with the

  1. Stock, S.P. Diversity, Biology and Evolutionary Relationships. In Nematode Pathogenesis of Insects and Other Pests; Campos-Herrera, R., Ed.; Springer International Publishing: Cham, 2015; pp. 3–27 ISBN 978-3-319-18265-0 978-3-319-18266-7.

Frequently cited ref. 22 (Clarke 2020) has also been limited.

Reviewer 2 Report

Comments and Suggestions for Authors

In this review, authors summarized the natural products derived from nematophilic bacteria, which is attractive and valuable leads for therapeutics. There are some tips to be revised.

1.      Line 19, “a nematode and an insect”, what kind of insect?

2.      Line 27, “such us”, should this revise into “such as”?

3.      Where the sequences of bacteria were found? Please supply the search method and if there have strain number of the searched bacteria.

4.      The structures of compounds in fig 2 are not clear?

5.      The NPs should be numbered to be more clearly and readable.

Comments on the Quality of English Language

Minor editing of English language required.

Author Response

We would like to thank the Reviewer for the useful comments to improve the paper. We have addressed all the comments as explained below.

Comment 1: “Line 19, “a nematode and an insect”, what kind of insect?”

Replay: It has been corrected. The term ‘a wide range of insect species” was used.

Comment 2: “ Line 27, “such us”, should this revise into “such as”?”

Replay: It has been corrected.

Comment 3: “Where the sequences of bacteria were found? Please supply the search method and if there have strain number of the searched bacteria.”

Replay: According Reviewer’s remark more  information to the description of Fig. 1 has been added “The sequences were retrieved from the GenBank using BLAST (NCBI). Multiple sequence alignment was created using ClustalW and the evolutionary distances were computed  using Maximum Composite Likelihood method in MEGA11”.

Comment 4:  “The structures of compounds in fig 2 are not clear?”

Replay: It has been corrected. The structures have been magnified, and the entire image has been inserted in horizontal orientation, making the details clearly visible.

Comment 5: “The NPs should be numbered to be more clearly and readable”

Replay: This comment has been taken into account. It was assumed that this is about the NPS located in Fig. 1. They were numbered to make them more readable.

Comment 6: “Minor editing of English language required.”

Replay: The text of the article was once again checked by an experienced English speaker. Several typos were caught, and, in addition, some sentences or parts of them were rephrased to make the language flawless. All language changes made are visible in the ‘track changes’ mode
